# Association between Children’s Engagement in Community Cultural Activities and Their Mental Health during the COVID-19 Pandemic: Results from A-CHILD Study

**DOI:** 10.3390/ijerph182413404

**Published:** 2021-12-20

**Authors:** Yui Yamaoka, Aya Isumi, Satomi Doi, Takeo Fujiwara

**Affiliations:** 1Department of Global Health Promotion, Tokyo Medical and Dental University, Tokyo 113-8519, Japan; yamaoka.hlth@tmd.ac.jp (Y.Y.); isumi.hlth@tmd.ac.jp (A.I.); doi.hlth@tmd.ac.jp (S.D.); 2Japan Society for the Promotion of Science, Tokyo 102-0083, Japan

**Keywords:** cultural engagement, community participation, community activity, child, mental health

## Abstract

Social learning experiences developed through engagement in community cultural activities can affect a child’s development. Few studies have examined how children’s engagement in community activities is related to their mental health. This study aimed to examine associations between children’s participation in community cultural activities and their mental health. We targeted all sixth-grade children in all 69 primary schools in Adachi City, Tokyo, using the Adachi Child Health Impact of Living Difficulty (A-CHILD) study (*n* = 4391). Parents answered the validated Japanese version of the Strength and Difficulties Questionnaire (SDQ) to assess child mental health, the child’s engagement in community cultural activities. The community activity in which children most frequently participated was local festivals. Participating in local festivals was significantly associated with lower behavioral difficulties (β = −0.49, SE = 0.17, *p* = 0.005) and higher prosocial behaviors (β = 0.25, SE = 0.07, *p* < 0.001) after adjusting for demographic variables, family social capital, and parent-child interactions. These results highlight the importance of children’s engagement in community cultural activities for their mental health during the COVID-19 pandemic.

## 1. Introduction

While a child’s development is mainly shaped by their parents during early childhood, as they grow older, children become more directly affected by extra-family contexts, such as their peers, school, neighbors, and community [1]. Neighborhood characteristics, including the physical, social, and educational environment, have been reported to have positive effects on child health, development, and wellbeing [2,3]. Higher social capital is associated with a lower risk of infant abuse [4] and behavioral problems [5]. Further, some community activities can boost social capital [6,7], including community cultural activities, which are considered one of the most promising events for doing so because engagement is also beneficial for reducing the risk of depression or anxiety among older adults [8]. This may be because participating in local festivals, which highlight shared local values and culture, is related to higher subjective wellbeing [9]. However, to our knowledge, the impact of cultural engagement has mostly been studied in adults and the aging population [10], while no study has yet investigated the impact on child mental health.

The 2019 coronavirus disease (COVID-19) pandemic has made a dramatic impact on all people, including children. Public health measures such as lockdowns, quarantines, school closures, and social distancing have been widely implemented around the world [11]. Several studies have reported that these circumstances of the COVID-19 pandemic are having a negative impact on mental health in the general population [12,13], including children [14,15]. Lockdowns, in which people are confined to their homes, are associated with less physical activity and green exposure [16], deceased social activity (i.e., visiting families and neighbors, going to places of entertainment, participating in community activities), and lower life satisfaction among the general population [17]. While studies have suggested the importance of promoting positive parenting to maintain child mental health [18,19,20,21], it remains unclear how engagement in community cultural activities impacts child mental health after accounting for family relationships and parent-child interactions during the COVID-19 pandemic. Japan is an attractive country in which to investigate the association between community cultural activities and child mental health because the nation has not been placed under any strict lockdown. Instead, whether or not community cultural activities continue to be held is at the discretion of local communities, and whether or not children participate in these activities is at the discretion of the children and their family members.

This study aimed to examine the association between community engagement and child mental health among Japanese children during the COVID-19 pandemic after adjusting for parent-child relationship-related factors.

## 2. Materials and Methods

### 2.1. Data Collection and Study Subjects

We used data from the Adachi Child Health Impact of Living Difficulty (A-CHILD) study [22]. A-CHILD aimed to examine the circumstances of children living in poverty and their physical and mental health in all primary schools located in Adachi City. Adachi City is one of the 23 special wards in Tokyo Prefecture, with approximately 690,000 population. As part of the A-CHILD study, a questionnaire was distributed to all sixth-grade children (aged 11–12 years old) in all 69 primary schools (*n* = 5355). The questionnaires were distributed from each school, and parents of targeted children completed and returned through the child’s school in an anonymous sealed envelope (response rate = 83.8%, *n* = 4489). The questionnaires included the demographics of the households, parents, and children, parenting behaviors, the psycho-social circumstances, and lifestyle behaviors of the child and parents. The details of survey items were explained elsewhere [22]. In Japan, schools were closed as part of a stage of emergency implemented during the first wave of the pandemic from April to May 2020. The current survey was conducted in October 2020 between the second (August) and third (December–January 2021) waves of the COVID-19 pandemic in Japan, when a voluntary social distancing policy was recommended in the absence of school closures, a state of emergency, and restrictions to travel. Our study included 4391 participants (82.0% of the target population) after excluding those with missing outcome variables (from the Strength and Difficulties Questionnaire). This study was approved by the Institutional Review Board of Tokyo Medical and Dental University (M2016-284-02).

### 2.2. Children’s Engagement in Community Cultural Activities

Parents responded to questions about whether their child had participated in community cultural activities during the past year. Participation in four types of activities was assessed: (1) local festivals (called “*Omatsuri*”) in the neighborhood, (2) seasonal events organized by child associations or neighborhood associations, (3) community cleaning or evacuation training events, and (4) classes at childcare centers or community learning centers. Parents provided a dichotomized answer (yes/no) on their child’s participation in the four types of activities. Local festivals, particularly summer festivals, are popular and culturally and traditionally embedded in the community [23]. For example, “*Bon-odori”* is a cultural dancing event that forms part of many summer festivals held in the middle of August to commemorate one’s ancestors. Child associations and neighborhood associations (“*Kodomokai*” or “*Chounaikai*”) are autonomous organizations established by residents for the purpose of promoting welfare, culture, and living conditions in the community. These organizations host seasonal and cultural events such as “*Undokai*” (sports festival) in the fall and “*Mochitsuki*” (pounding rice cake) during the new year season as rice cake is traditionally eaten over the new year in Japan. Cleaning activities and evacuation training events are voluntary activities that are also widely conducted in communities. Volunteer residents gather to help clean up local streets or parks. Further, because of the frequency of natural disasters such as earthquakes, Japanese children often receive evacuation training at school and sometimes in the community. Moreover, childcare centers (“*Jidoukan*”) and community learning centers in the community often have facilities such as playrooms and a library in which they provide informal educational classes to residents for little to no fee. Although cleaning and learning opportunities may not necessarily represent Japanese culture, we included these activities as community cultural activities because the same local community groups (“*Jichikai*” or “*Chounaikai*”) plan and manage these activities for children to raise a sense of community.

### 2.3. Child Mental Health

This study used the validated Japanese version of the Strength and Difficulties Questionnaire (SDQ), which is widely used as a brief screening tool for child mental health [24]. SDQ has been translated into more than 80 languages. The Japanese versions of the parent and teacher-reported SDQ have shown satisfactory internal consistency (α = 0.81) and convergent validity [24]. SDQ was widely used to assess child mental health in other COVID-19 studies (e.g., Ravens-Sieberer et al. [25] and Wang et al. [26]). In the A-CHILD survey, parents responded to 25 items inquiring about their child’s behavior during the past six months on a three-item scale (2 = very true, 1 = true, and 0 = not true). The SDQ includes four subscales on behavioral problems (emotional problems, conduct problems, hyperactive/inattention problems, peer relationship problems) and one subscale on prosocial behavior. Each subscale consists of five items each (range: 0–10). The SDQ total difficulties score is obtained by summing the scores from the four subscales on behavioral problems (range: 0–40) and represents the degree of behavioral difficulties in children. Higher SDQ total scores indicate greater problematic behaviors, and higher prosocial scores indicate better social skills in children. The recommended cut-off ranges of total difficulties score are 0–12 for normal, 13–15 for borderline, and 16–40 for clinical range. For prosocial behavior, the cut-off ranges are 6–10 for normal, 5 for borderline, and 0–4 for clinical range [27].

### 2.4. Family Relationship

We used the family social capital (SC) score developed previously [28] to assess family relationships with some modifications to focus on family cohesion, trust, and reciprocity. The family SC score includes seven items: (1) I like my family; (2) I enjoy spending time with my family; (3) I consult with my family when in trouble; (4) I trust my family; (5) my family goes out during summer vacations and celebrates birthdays of family members; (6) my family follows family rules; and (7) my family helps each other when in trouble. Parents responded on a 5-point Likert scale (4 = strongly agree, 3 = agree, 2 = neither, 1 = disagree, and 0 = strongly disagree). The Cronbach’s alpha of the seven items was 0.88. The total family SC score (range: 0–28) was categorized into three groups based on its distribution: high (score = 28, 30.7%), middle (score = 24–27, 38.3%) and low (score ≤ 23, 30.4%). The higher categories indicate a better family relationship.

### 2.5. Parent-Child Interactions

The survey inquired about parental involvement in daily activities at home. The following eight items were assessed: (1) watching the child study, (2) playing card games or board games, etc. with the child, (3) talking about school with the child, (4) talking about the future or career path with the child, (5) talking about news on politics and socioeconomic and social issues with the child, (6) talking about TV programs with the child, (7) cooking with the child, and (8) going out with the child. Parents indicated the frequency of each interaction on a 5-point Likert scale (4 = everyday, 3 = 3–4 times per week, 2 = 1–2 times per week, 1 = 1–2 times per month, and 0 = rarely). As pairwise correlations ranged from 0.078 to 0.470 among the eight interactions, all parent-child interactions were used in the analysis. A higher score indicates more interaction with the child.

### 2.6. Covariates

We used the following demographic variables assessed in the questionnaire: child’s age in months and sex, number of parents in the household (one or two), number of children in the household (one, two, three, four or more), maternal age (<35, 35–39, 40–44, ≥45 years old), and low income. We defined low income as an annual household income of below 3.0 million yen (approximately 2800 USD) based on the definition of living with low income used in a previous study [22]. No response to the question on income was used as a dummy variable (13.4%).

### 2.7. Statistical Analysis

First, we extracted the demographic variables and data on children’s engagement in community cultural activities, family SC score, parent-child interactions, and SDQ score. Second, we performed linear regression analyses to examine the relationship between community engagement and two outcomes (SDQ total score and prosocial score) in Model 1. We added demographic variables to Model 2. Subsequently, we added either family SC score as a categorical variable (Model 3) or the scores of the eight items used to assess parent-child interactions simultaneously (Model 4). The analyses were conducted using Stata/MP version 16.1 (STATA Corp., College Station, TX, USA).

## 3. Results

Table 1 shows the demographic variables, the proportion of children who engaged in cultural activities, and family SC and parent-child interaction scores. Among 4391 children, half were male, and over one-third of mothers were 45 years old and older. A total of 14% of participants were single parents, and nearly half of the parent(s) had two children. A total of 1 in 10 children lived in low-income families. The parent-child interaction children and their parents most frequently engaged in was talking about school (mean = 3.43, SD = 0.89) followed by talking about TV programs (mean = 2.78, SD = 1.14). The community cultural activity in which children participated most frequently was local festivals in the neighborhood (53.0%), followed by seasonal events organized by child associations and neighborhood associations (24.9%). Approximately two-thirds of children (63.2%) participated in at least one type of community activity during the past year.

Table 2 shows the mental health of children based on parent-reported SDQ. Among children in sixth grade, 11.3% of them showed a clinical range of great behavioral difficulties, and 16.7% of them showed a clinical range of low prosocial behaviors.

Table 3 shows the associations between community engagement and SDQ total score. Hereafter, we refer to the four types of activities as ‘festivals’, ‘events’, ‘cleaning’, and ‘classes’. In Model 1, participation in both festivals and classes was significantly associated with lower behavioral difficulties. After adding demographic variables, participation in festivals remained significantly associated with lower behavioral difficulties, while participation in classes showed no significant association. Even after adding family SC and parent-child interaction scores, participation in festivals remained significantly associated with lower behavioral difficulties (Model 3: β = −0.33, *p* = 0.046; Model 4: β = −0.42, *p* = 0.014).

Table 4 shows the associations between community engagement and SDQ prosocial score. In Model 1, engagement in three types of community activities, festivals, cleaning, and classes, was significantly associated with higher prosocial behaviors. After adding family SC scores, engagement in festivals and cleaning remained significantly associated with higher prosocial behaviors (festival: β = 0.19, *p* = 0.004; cleaning: β = 0.21, *p* = 0.032). After further adding parent-child interaction scores, cleaning was no longer significantly associated with prosocial behavior, while festivals remained significantly positively associated with prosocial behaviors (β = 0.21, *p* = 0.032).

## 4. Discussion

The current study reported child mental health during the COVID−19 pandemic among all sixth-grade children from all primary schools in Adachi City, Tokyo. Using population-based samples, we found that, during the pandemic, 11.3% of children showed the clinical range of total difficulties score, and the mean score was 8.7. Compared to the previous studies in pre-pandemic in Japan, the percentage of the clinical range was 9.5% among children aged 4–12 years old [27], and the mean score was 7.2 of children aged 10–12 years old from nationwide community samples using parent-reported SDQ score [24]. Therefore, our samples in this study showed a higher percentage of behavioral problems. In terms of prosocial behavior, our study showed greater prosocial problems during the pandemic than pre-pandemic. We found, during the pandemic, 16.7% of children showed the clinical range of prosocial behavior score, and the mean score was 6.5, while, pre-pandemic, the percentage of clinical range in prosocial behavior was 13.3% among children aged 4–12 years old [27], and the mean score was 6.3 of children aged 10–12 years old [24]. The percentage of children with a clinical range in prosocial behavior in the current study was slightly higher than normative samples, and the proportion of clinical range was also higher. It indicated that most children might lose the opportunity to improve their prosocial skills during the pandemic.

One recent study reported child mental health problems during the pandemic in Japan. The study compared child’s mental health problems at two time points during the pandemic (wave 1: March 2020, wave 2: May 2020) and reported that clinical-level problems were greater in wave 2 in all subdomains of SDQ [29]. The percentages of clinical range for waves 1 and 2 were 23.8% and 30.0%, which were higher than our findings. The possible reasons were differences in study period and population. The study conducted surveys at the beginning of the COVID-19 pandemic and the school closure period, and their employed online survey; therefore, the more motivated parents regarding child mental health problems might participate in the survey. The current study was conducted between the second (August 2020) and third (December–January 2021) waves of the COVID-19 pandemic in Japan, using a population-based sample, and the school closures were not in place at that time.

Social participation is an essential opportunity for human development. Most studies on social participation to date have targeted older adults [30,31], children with disability [32], or vulnerable populations such as refugees [33]. This study focused on the social participation of a general child population, particularly in community-based cultural activities such as local festivals and seasonal events, cleaning and evacuation training events, and classes held at community centers. Our findings revealed that approximately two-thirds of children participated in at least one type of community cultural activity during the COVID-19 pandemic in Japan. Most children in sixth grade, the final year of elementary school, grow up in the community within their school’s district. Our results indicate that community cultural activities play an important role in the development of children’s mental health.

The current study revealed significant independent associations between participation in local festivals and favorable mental health among children even after controlling for family SC and parent-child interactions. Local festivals and events foster a sense of community, which strengthens a person’s attachment to their place of residence by highlighting values and culture, and providing a means for connection and belonging and social inclusion [34]. While cleaning is seen as a job in Western countries, Japanese society considers cleaning a form of character development [35]. Thus, cleaning is incorporated into school curricula and local activities. Evacuation training is also embedded with Japan’s “Disaster culture,” which throughout history has developed community capacity and resilience [36]. Further, classes conducted at childcare and community learning centers are wide ranging and can include cultural arts and music for individuals of all ages. These community classes are promoted by the nation’s policy for lifelong learning [37]. Participation in these community-based cultural activities may have protective effects on children’s mental health, possibly by fostering a sense of community and understanding of values and culture and allowing participants to build stronger networks with their peers and neighbors. A previous study reported that there was no link between participating in global cultural activities such as cultural festivals and feeling anxious or unhappy, which suggests that local festivals have greater potential to encourage even people with poor mental health to participate [38]. Given that local festivals were the most popular event among children aged 11–12 years old in this study, such festivals have the potential to play an essential role in promoting the social inclusion of all children with or without special needs or adversities. Further research is needed to understand the factors affecting social participation, including barriers, promoters, and those causing children to be left behind.

### Limitations and Further Directions

There were several limitations in this study. First, because this was a cross-sectional study, we cannot rule out the possibility of a reverse association; that is, children may have participated in local activities because they have better mental health. Second, we did not examine associations between changes in participation in community activities and changes in mental health to account for unmeasured participant-related confounders such as genetic factors or personality. Third, child mental health was assessed by their parents in this study. The A-CHILD survey did not include a child-reported version of SDQ due to the limitation of the additional volume of items. One recent study compared child-parent agreements on SDQ in Japanese adolescents and reported that adolescents rated higher scores in total difficulties than their parents [39]. Further research is needed to assess child mental health from multiple informants, such as adolescents themselves or teachers. Finally, because our study participants were limited to one city in Tokyo, our results may not be generalizable to other parts of Japan, such as rural areas where cultural traditions are typically more strongly rooted. Further research is needed to examine the association between different types of community-based cultural activities and child mental health across Japan.

## 5. Conclusions

This study showed that children’s engagement in community cultural activities is independently associated with lower behavioral difficulties and higher prosocial skills, regardless of parent-child relationships during the COVID-19 pandemic. Further research and policies are needed to identify ways to provide children with social experiences while infection control measures are being implemented during a global pandemic.

## Figures and Tables

**Table 1 ijerph-18-13404-t001:** Demographic and other variables of interest.

		(Range)	Mean/N	SD/%
Demographics	Child age (month)	(139–150)	144	3.33
	Child sex	Female	2157	49.1%
		Male	2234	50.9%
	Maternal age	45+	1723	39.2%
		40–44	1462	33.3%
		35–39	814	18.5%
		<35	250	5.7%
		Missing	142	3.2%
	Number of parents	Two	3777	86.0%
		Single	614	14.0%
	Number of children	One	855	19.5%
		Two	2163	49.3%
		Three	1070	24.4%
		Four+	303	6.9%
	Income of household	3M+	3367	76.7%
		<3M	456	10.4%
		Missing	568	12.9%
Family relationship	Family social capital score	High	1345	30.6%
		Middle	1697	38.6%
		Low	1340	30.5%
		Missing	9	0.2%
Parent-child interactions	Watch child study	(0–4)	2.00	1.41
	Play with child	(0–4)	0.66	0.84
	Talk about school	(0–4)	3.43	0.89
	Talk about future	(0–4)	1.90	1.22
	Talk about news	(0–4)	2.05	1.28
	Talk about TV program	(0–4)	2.78	1.14
	Cook with child	(0–4)	0.98	0.95
	Go out with child	(0–4)	1.82	0.86
Participation for community activities during last year	Festivals in neighborhood	Yes	2332	53.1%
		No	2059	46.9%
	Events for children and neighborhood association	Yes	1106	25.2%
		No	3285	74.8%
	Community cleaning, evacuation training	Yes	560	12.8%
		No	3831	87.2%
	Classes at childcare centers and community learning centers	Yes	738	16.8%
		No	3653	83.2%

**Table 2 ijerph-18-13404-t002:** Child emotional and behavioral problems from parent-reported Strength and Difficulties Questionnaire.

		Mean	SD
Total difficulties score	(Range: 0–40)	8.73	5.31
Emotional problems	(Range: 0–10)	1.71	1.84
Conduct problems	(Range: 0–10)	2.13	1.77
Hyperactivity problems	(Range: 0–10)	2.86	2.14
Peer problems	(Range: 0–10)	2.04	1.78
Prosocial behavior	(Range: 0–10)	6.53	2.13
		N	%
Total difficulties score	Normal (0–12)	3453	78.6%
	Borderline (13–15)	441	10.0%
	Clinical (16–40)	497	11.3%
Prosocial behavior	Normal (6–10)	2951	67.2%
	Borderline (5)	707	16.1%
	Clinical (0–4)	733	16.7%

**Table 3 ijerph-18-13404-t003:** Associations between participation in community activities and SDQ total score.

	Model 1 (Univariate)			Model 2 (Adjusted)			Model 3 (Adjusted)			Model 4 (Adjusted)		
	Coef	>s.e.	*p*-Value	Coef	s.e.	*p*-Value	Coef	s.e.	*p*-Value	Coef	s.e.	*p*-Value
Participation in community activities												
Festivals in neighborhood	−0.57	0.17	0.001	−0.49	0.17	0.005	−0.33	0.17	0.046	−0.42	0.17	0.014
Events for children and neighborhood association	−0.37	0.20	0.067	−0.34	0.20	0.089	−0.30	0.19	0.123	−0.28	0.20	0.163
Community cleaning, evacuation training	−0.08	0.26	0.761	0.05	0.25	0.854	0.14	0.24	0.564	0.11	0.25	0.659
Classes at childcare centers and community learning centers	−0.55	0.22	0.013	−0.42	0.22	0.057	−0.41	0.21	0.056	−0.20	0.22	0.360
Family social capital (ref: High)												
Middle							1.73	0.18	<0.001			
Low							3.81	0.20	<0.001			
Parent-child interactions												
Watch child study										−0.12	0.06	0.044
Play with child										0.05	0.10	0.636
Talk about school										−0.75	0.10	<0.001
Talk about future										0.04	0.08	0.574
Talk about news										−0.23	0.08	0.003
Talk about TV program										−0.16	0.08	0.055
Cook with child										−0.11	0.09	0.227
Go out with child										−0.14	0.10	0.174

Model 2, 3, 4 adjusted covariates (child sex, age, maternal age, number of children, and low income).

**Table 4 ijerph-18-13404-t004:** Associations between participation in community activities and SDQ prosocial score.

	Model 1 (Univariate)			Model 2 (Adjusted)			Model 3 (Adjusted)			Model 4 (Adjusted)		
	Coef	s.e.	*p*-Value	Coef	s.e.	*p*-Value	Coef	s.e.	*p*-Value	Coef	s.e.	*p*-Value
Participation in community activities												
Festivals in neighborhood	0.26	0.07	<0.001	0.25	0.07	<0.001	0.19	0.07	0.004	0.19	0.07	0.004
Events for children and neighborhood association	0.07	0.08	0.357	0.02	0.08	0.815	0.00	0.08	0.997	−0.03	0.08	0.709
Community cleaning, evacuation training	0.27	0.10	0.008	0.25	0.10	0.015	0.21	0.10	0.032	0.18	0.10	0.071
Classes at childcare centers and community learning centers	0.19	0.09	0.030	0.13	0.09	0.143	0.12	0.09	0.150	0.00	0.09	0.978
Family social capital (ref: High)												
Middle							−0.79	0.07	<0.001			
Low							−1.53	0.08	<0.001			
Parent-child interactions												
Watch child study										−0.02	0.02	0.392
Play with child										0.06	0.04	0.127
Talk about school										0.29	0.04	<0.001
Talk about future										0.09	0.03	0.002
Talk about news										0.07	0.03	0.031
Talk about TV program										0.07	0.03	0.043
Cook with child										0.32	0.04	<0.001
Go out with child										0.14	0.04	<0.001

Model 2, 3, 4 adjusted covariates (child sex, age, maternal age, number of children, and low income).

## Data Availability

Data sharing is not applicable to this article.

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
