# Peer review of "Association between Children’s Engagement in Community Cultural Activities and Their Mental Health during the COVID-19 Pandemic: Results from A-CHILD Study"

_ijerph, 2021, doi:10.3390/ijerph182413404_

Round 1
Reviewer 1 Report
The state of mental health has been repeatedly studied during the COVID pandemic, and the reviewer agrees that the focus here is mostly on adults (including the elderly). However, when reading the manuscript, several important facts were noticed.
The reviewer considers the title of the article to be misleading. Using the chosen method (questionnaire and only the parents' opinion based on the questionnaire) does not allow to say that it is the mental health of the children. If only the opinion of the parents with the questionnaire is examined, then it is only the opinion of one of the parties. This should be stated honestly and unequivocally, both in the title and in the debate.- Why were children not asked directly about mental health indicators (behavioral difficulties and prosocial skills)? In such a case, it would have been possible to at least compare the parents' opinion and the child's self-esteem.
- Is it known how many mental health problems were diagnosed in this age group in Tokyo before COVID and during the COVID pandemic? Are there any diagnostic or statistical differences?
- Are there analogous data (behavioral difficulties and prosocial behavior) for the period before COVID? How could the situation be different from pre-COVID? The discussion does not provide further information on this.
Other comments:
- Supplement the abstract with a description of the methodology.
- The formatting of Tables 1 and 2 is revised to give a long and dragging impression.
Best wishes!
Author Response
Thank you for kindly reviewing our manuscript and valuable comments. We agree with the importance of focusing on mental health among children. Please see point-by-point responses and the revised manuscript based on reviewers’ comments.

Reviewer 2 Report
The manuscript is of interest and overall well-written. There are few minor issues which need to be addressed:
- Please state in the Methods if the questionnaire was pre-tested on a sample not part of the final study population.
- Was sample size estimated prior to study beginning?
- Please add as an Appendix the questionnaire employed.
Author Response
Thank you for kindly reviewing our manuscript and valuable comments. Please see point-by-point responses and the revised manuscript based on reviewers’ comments.

Reviewer 3 Report
This is a well-written interesting article on the association between children’s engagement in different cultural activities in Tokyo, Japan and their mental health during the current pandemic. The work is of interest for the Journals' readership and in my opinion can be accepted after minor revisions:
-less physical activity, especially outdoors, is another issue to be mentioned briefly in the Introduction after the mention of the decreased social activity, for example. cite for example work with DOI : 10.1007/s10311-021-01321-9
-line 80: Bon-odori should be in italics within brackets
-lines 113-116: use ';' in place of ',' and lowercase instead of capital letters in the list. For example: 'family; 5) my' in place of 'family, 5) My' etc
-separate with blank spaces the frame bewtween lines 162 and 170 and the upper Table as it looks as the legend of Tab 1 in the current form
-check that all numbers mentioned in the manuscript text are consistent with those reported in the Tables as I found some problems for example with 'SD=0.98' (text, line 159) and Table 1
-Tables 2 and 3: writings on the left side should be separated each other to make the tables clearer to the readers. For example leave a white space between 'Local festivals in the neighborhood' and 'Events organized
by child or neighborhood associations' . This should be done for all writings of both the tables.
-lines 172-173 and 183-184 report the same explaination. Report this just once in Materials and methods indicating that it refers to both tables.
-line 191: check 'at held at': may be 'held at'?
-lines 225-228: thera are words separated in two (commu-nity; partici-pant; general-izable): join them.
Author Response
Thank you for kindly reviewing our manuscript and valuable comments. We modified the manuscript with point-by-point responses as below.

Round 2
Reviewer 1 Report
Dear authors,
you have substantially corrected the deficiencies and supplemented the manuscript. I am sure your article will be interesting and informative to many, and I wish you a peaceful and beautiful Christmas.Reviewer 2 Report
The issues addressed in the previous review were solved. There are no further comments.